# The Gut Microbiome, Seleno-Compounds, and Acute Myocardial Infarction

**DOI:** 10.3390/jcm11051462

**Published:** 2022-03-07

**Authors:** Fu-Chun Chiu, Chin-Feng Tsai, Pang-Shuo Huang, Ching-Yu Shih, Mong-Hsun Tsai, Juey-Jen Hwang, Yi-Chih Wang, Eric Y. Chuang, Chia-Ti Tsai, Sheng-Nan Chang

**Affiliations:** 1Division of Cardiology, Department of Internal Medicine, National Taiwan University Hospital Yun-Lin Branch, Dou-Liu City 640, Taiwan; uenling@hotmail.com (F.-C.C.); b93401021@gmail.com (P.-S.H.); jueyhwang@ntu.edu.tw (J.-J.H.); 2Division of Cardiology, Department of Internal Medicine, Chung Shan Medical University Hospital, Taichung City 402, Taiwan; alberttsai54@gmail.com; 3School of Medicine, Chung Shan Medical University, Taichung City 401, Taiwan; 4Bioinformatics & Biostatistics Core Lab, Centers of Genomic and Precision Medicine, National Taiwan University, Taipei City 100, Taiwan; cyshih30@gmail.com (C.-Y.S.); motiont@gmail.com (M.-H.T.); 5Institute of Biotechnology, National Taiwan University, Taipei City 100, Taiwan; 6Division of Cardiology, Department of Internal Medicine, National Taiwan University Hospital, Taipei City 100, Taiwan; med011@seed.net.tw (Y.-C.W.); cttsai1999@gmail.com (C.-T.T.); 7Graduate Institute of Biomedical Electronics and Bioinformatics, National Taiwan University, Taipei City 100, Taiwan; chuangey@ntu.edu.tw

**Keywords:** myocardial infarction, acute coronary syndrome, microbiota, *Selenomonadales*, seleno-compound

## Abstract

Background: Gut microbiome alterations might be considered a metabolic disorder. However, the relationship between the microbiome and acute myocardial infarction (AMI) has not been properly validated. Methods: The feces of 44 subjects (AMI: 19; control: 25) were collected for fecal genomic DNA extraction. The variable region V3–V4 of the 16S rRNA gene was sequenced using the Illumina MiSeq platform. The metabolite amounts were analyzed using the Kyoto Encyclopedia of Genes and Genomes (KEGG) metabolic pathways. Results: The bacteria were more enriched in the AMI group both in the observed operational taxonomic units (OTUs) and faith phylogenetic diversity (PD) (*p*-value = 0.01 and <0.001 with 95% CI, individually). The *Selenomonadales* were less enriched in the AMI group at the family, genus, and species levels (all linear discriminant analysis (LDA) scores > 2). Seleno-compounds were more abundant in the AMI group at the family, genus, and species levels (all LDA scores > 2). Conclusions: This is the first study to demonstrate the association of *Selenomonadales* and seleno-compounds with the occurrence of AMI. Our findings provide an opportunity to identify a novel approach to prevent and treat AMI.

## 1. Introduction

Acute myocardial infarction (AMI) is a growing epidemic in developing countries and is the leading cause of death in industrialized societies. Currently, the risk factors associated with AMI investigation are based on the inflammation and cholesterol bio-synthesis associated with hypertension, diabetes mellitus, dyslipidemia, smoking, and the post-menopause period [1]. However, these factors have already been well investigated and there is a need to identify another candidate.

Alteration in the gut microbiome (dysbiosis) is associated with diseases such as obesity, diabetes, dyslipidemia, malignancies, autoimmune disorders, and cardiovascular diseases [2]. There are more than 2000 species of bacterial organisms in the human body, and the gut microbiome is by far the greatest mass of microbiota [3]. These bacterial organisms have developed complicated connections with the human body. Previous studies have shown that the gut microbiome can cause cardiovascular diseases in multiple ways [4]. For instance, the gut microbiome influences the activity of protein disulfide isomerase, involving the protein-folding process associated with inflammation [5]. Adenosine triphosphate (ATP) production and ketone body metabolism by anaerobic succinate formation are also regulated by the gut microbiome under ischemia [2,6]. Gut microbiome-dependent trimethylamine-N-oxide (TMAO) production has links to alterations in macrophage and endothelial cell phenotypes, promotion of platelet hyperresponsiveness, coronary atherosclerotic burden, plaque complexity, acute coronary syndrome, and post-myocardial infarction sequelae [7,8,9]. Moreover, intestinal dysbiosis induced by circulatory antibiotics and other intestinal microbial metabolites of amino acids (e.g., phenylalanine, tryptophan, and tyrosine) are associated with the severity of myocardial infarction in rats [2,10].

Therefore, the gut microbiome contributes to inflammation, energy production, and immune modulation and is possibly involved in AMI episodes. Accordingly, and given the need for the effective prevention and treatment of AMI, we conducted this comparative study involving 16S ribosomal RNA (rRNA) microbiome analysis to investigate the composition and functional differences between patients with and those without AMI.

The purpose of this research was to pinpoint the gut microbiome associated with AMI episodes. This new biological axis can help us better understand the pathophysiology of AMI and improve current AMI therapy.

## 2. Materials and Methods

### 2.1. Patient and Public Involvement Statement

This study was approved by the ethics committee and Institutional Review Board (IRB) on Human Research of the Medical Research Department of National Taiwan University Hospital, Taipei, Taiwan (IRB: 202005040RIND). All subjects provided their written informed consent to participate in this study. All research was performed in accordance with relevant guidelines/regulations. Research involving human research participants was performed in accordance with the Declaration of Helsinki.

### 2.2. Study Setting and Participants

In this single-center and case-control study, 360 patients were enrolled initially between August 2020 and October 2020. Of these, 185 patients with suspected acute coronary syndrome were assigned to the AMI group, and 175 patients were enrolled as the control group. Those who were included were (1) patients older than 20 years and (2) patients with suspected acute coronary syndrome, who were assigned to the AMI group. Those who were excluded were subjects (1) who had had prior gastrointestinal surgery (e.g., colectomy, ileectomy, and gastrectomy), (2) who had an ongoing infection and were currently on antibiotics, (3) who had inflammatory bowel disease, (4) who had auto-immune diseases, (5) who had malignancy, (6) with a history of stroke, (7) suffering from renal failure, (8) who had hepatic diseases (e.g., hepatitis B or were a hepatitis C carrier), (9) who had digestive diseases, (10) who habitually smoked or drank alcohol, (11) who had used antacids or probiotics within 3 months of fecal sample collection, and (12) of the AMI group without the final diagnosis of ST-elevation myocardial infarction (MI) after coronary artery intervention. The gastrointestinal functions of these subjects were normal, with no vomiting, diarrhea, and/or constipation up till the day of stool collection.

The blood pressure was acquired at initial enrollment and was the average of two consecutive readings 1 min apart obtained by OMRON 8712 devices (OMRON, HealthCare, Taipei City, Taiwan), with the participants in the seated position. All of the participants were provided a hospital diet and had not used any antibiotics for the preceding 3 months, the goal being to minimize any potential confounding effects on the microbiota. All of the AMI patients were diagnosed with ST-elevation MI, had accepted percutaneous coronary intervention, and were following medical treatment at the intensive care unit after coronary intervention according to the guideline of the American College of Cardiology [11]. No complications such as cardiogenic shock, hypotension, pulmonary congestion, and renal or respiratory failure were noted during hospitalization. Finally, 44 patients (AMI: 19; control: 25) remained for further analysis. A flow chart of the research is provided in Figure 1.

### 2.3. Fecal Collection and Processing

To minimize the potential confounding effects of antibiotics on the microbiota, it was ensured that none of the participants were given any antibiotics for at least 3 months after AMI or enrollment and the feces were collected at least 3 months after AMI. A sample of the first defecation in the morning was collected from each subject. The fecal samples were collected in sterile cryotubes (Iron Will Biomedical Technology, Taipei City, Taiwan). All samples were frozen in liquid nitrogen within 30 min. They were sealed and stored at −80 °C in freezers. Bacterial genomic DNA was isolated within 2 weeks of collection. The inner parts of the samples were used for sequencing to avoid environmental contamination. The DNA was extracted by using the stool DNA Kit (Omega Biotek, Norcross, GA, USA) according to the manufacturer’s protocol.

### 2.4. DNA Extraction

The DNA was extracted as follows: to a 2 mL microcentrifuge tube containing 200 mg of Glass Beads X, 200 mg of a stool sample was added. To this, 540 μL of SLX-Mlus buffer was added. The mixture was vortexed at the maximum speed for 10 min. To the mixture, 60 μL of DS buffer and 20 μL of proteinase K solution was added. Next, 200 μL of SP2 buffer was added. The mixture was vortexed at the maximum speed for 30 s. Then, the mixture was centrifuged at the maximum speed (≥13,000× *g*) for 5 min. Next, 400 μL of the supernatant was aspirated to a 1.5 mL microcentrifuge tube. To the mixture, 200 μL of cHTR reagent was added. The resulting mixture was vortexed at the maximum speed for 10 s. Then, 250 μL of the supernatant was transferred to a 1.5 mL microcentrifuge tube. To this, 250 μL of BL buffer and 250 μL of 100% ethanol was added. Then, a HiBind DNA Mini Column was inserted into a 2 mL collection tube. The mixture was centrifuged at the maximum speed for 1 min. The filtrate was discarded. To the remaining mixture, 500 μL of VHB buffer was added. This was centrifuged at the maximum speed for 1 min. The filtrate was discarded. Finally, the DNA was stored at −20 °C.

### 2.5. 16S rRNA Gene Sequencing

To investigate the compositional change in the microbiome associated with AMI, PCR amplification was performed on the V3–V4 region of the 16S rRNA gene with TransStart Fastpfu DNA polymerase (Takara Bio, San Jose City, CA, USA), followed by sequencing on the Illumina MiSeq v3 chemistry (Illumina Inc., San Diego, CA, USA) in multiple runs and pooling together of all 44 samples according to the manufacturer’s instructions. 

### 2.6. Comparison of the Gut Microbiome Composition

The raw 16S data were analyzed by PANDAseq, processed through the QIIME (version 1.8.0), and clustered into operational taxonomic units (OTUs) with a 97% identity cutoff. The alpha-diversity measures were calculated based on the OTU counts. The number of observed OTUs indicates microbial richness, which measures the number of taxa in each sample. The linear discriminant analysis effect size (LEfSe) was calculated using the online version of Galaxy 3. Linear discriminate analysis (LDA) was performed using a one-against-all strategy, and OTUs showing a score higher than 2 were selected. The Kyoto Encyclopedia of Genes and Genomes (KEGG) pathways database was used to predict the differences in the bacterial biochemical pathways between the patients with and those without AMI.

### 2.7. Statistical Analyses

Statistical analyses were performed by a researcher blinded to the subjects’ conditions. Continuous variables were expressed as the mean ± the standard deviation and were compared with the two-tailed Student’s *t*-test. The categorical variables were reported as frequencies or percentages. The non-normal variables were compared with the Mann–Whitney U test. We applied metagenomic sequencing for intestinal flora analyses, which included quality control; assembly; gene prediction; construction of nonredundant gene sets; gene function annotation; and quantitative abundance at the family, genus, species, and functional levels. The flora’s composition, diversity, multidimensional difference, function, and metabolites were also analyzed using the KEGG database. A two-tailed *p*-value < 0.05 was considered statistically significant. The statistical analyses were performed by the SPSS version 25.0 statistical software (SPSS, Chicago, IL, USA). 

## 3. Results

### 3.1. Basic Demographics

For the final analysis, 44 participants (AMI 19 (43.18%) vs. control 25 (56.81%); mean age: 65.15 ± 11.53 vs. 73.20 ± 9.06 years) were eligible (Table 1). In general, the AMI patients presented higher body weight and body height values than the controls (*p* < 0.001; 95% CI). The blood pressure measurements were not different between the AMI and control groups (Table 1). Overall, the AMI patients showed higher levels of glutamic oxaloacetic transaminase (GOT), glutamic pyruvic transaminase (GPT), cardiac enzyme (creatine kinase (CK)), and inflammation marker (C-reactive protein (CRP)) than the controls (Table 1). This difference was possibly due to AMI events.

### 3.2. Analysis of Diversity between Groups

In this study, the DNAs of the gut microbiome were extracted from the fecal samples, and the DNA libraries were constructed by 16S rRNA gene sequencing on the MiSeq (Illumina) platform. Highly similar sequences were grouped into OTUs. The alpha diversity was an indicator of species diversity. The intestinal microbial metagenomics analysis indicated a difference in the relative richness of the bacteria between the groups (Figure 2a). The bacteria were more enriched in the AMI group (*p* = 0.01; 95% CI) (Figure 2a). The faith phylogenetic diversity (PD) was often used as the indicator of feature diversity. As in the case of alpha diversity, the phylogenetic analog of taxon analyses revealed more richness of bacteria in the AMI group (*p* < 0.001; 95% CI) (Figure 2b). The uniformity analysis was performed in terms of Pielou’s evenness. There was no difference in the evenness between the groups (*p* = 0.4; 95% CI) (Figure 2c). The Shannon index, a statistic information index, assumed that all species were represented in a sample and that they were randomly sampled. The Shannon index between the groups was similar (*p* = 0.23; 95% CI) (Figure 2d). To sum up, the gut microbiome was more abundant in the AMI group (Figure 2a,b).

To explore these findings, we performed a LEfSe analysis to identify the different taxa between patients with and those without AMI. The taxonomic cladogram reported all clades of the gut microbiome, and there were different bacterial genera between the groups. *Selenomonadales* and *Lachnospiraceae* concomitantly increased in the control group at the family level (LDA > 2) (Figure 3a and Appendix A). The control group was also enriched with clusters of *Selenomonadales* at the genus and species levels (all LDA scores > 2) (Figure 3b,c, Appendix A).

Because the gut microbiome produced and consumed many metabolites, we assume that certain metabolites were associated with the abundance of the gut microbiome. The KEGG database resource was used to estimate the difference between the metabolic pathways of the groups. Functional metagenome analysis revealed a difference in the seleno-compound distribution between the groups. Seleno-compounds were more abundant in the AMI group at the family, genus, and species levels (all LDA scores > 2) (Figure 4a–c).

## 4. Discussion

In the present study, we explored the association between gut microbiome and AMI in patients. We made several novel findings. First, the bacteria were more enriched in the AMI group (*p* = 0.01; 95% CI). Second, *Selenomonadales* was more abundant in the patients without AMI (family, genus, and species levels; all LDA scores > 2). Third, the seleno-compounds were more plentiful in the AMI group (family, genus, and species levels; all LDA scores > 2).

### 4.1. Gut Microbiome Is Influenced in AMI Patients

AMI episodes are allegedly associated with a sequential change in the gut–brain axis, which substantially alters the gut microbiome (dysbiosis), influencing other organs, including the brain, after ischemia [4,12]. These inflammatory changes attributed to AMI might be due to a change in neutrophil expression. For instance, in [13], increasing amounts of adherent-activated neutrophils formed neutrophil extracellular traps and triggered an inflammatory reaction of the splanchnic circulation. These neutrophil extracellular traps injured the mesenteric venules under ischemic conditions [13]. Therefore, intestinal barrier dysfunction, bacterial translocation, and post-ischemic infection often occur with AMI [4]. In response to this, the composition of the gut microbiome also changes (dysbiosis) under an ischemic status. In [13], the gut microbiome was noted to suppress the hyper-reactivity of neutrophil extracellular traps by enhancing neutrophil recruitment in mice. Additionally, dysbiosis was found to suppress the effector T cells trafficking from the gut to the leptomeninges after a stroke by increasing regulatory T cells and reducing IL-17+ γδ T cells through altered dendritic cell activity [12]. To sum up, dysbiosis alters the inflammatory reaction and immune homeostasis under ischemic episodes [12].

Clinically, several species of gut microbiomes are associated with ischemic events. *Enterobacteriaceae* and *Streptococcus* spp. increased in abundance in subjects with atherosclerotic cardiovascular disease under a stable status [7]. Other studies have also declared an association between coronary artery disease and bacterial pathogens, such as *Helicobacter pylori* and *Chlamydia pneumonia*. [2] Moreover, a lower abundance of the phylum *Firmicutes* and a slightly higher abundance of the phylum *Bacteroidetes* were found in AMI patients [14]. Compared with these findings, there was less abundance of *Selenomonadales* in the AMI group relative to that in the control group at the family, genus, and species levels in the present study. Dietary habits or coronary artery disease might change these results.

### 4.2. Mechanisms Associated with Selenomonadales and AMI

This is the first study to demonstrate the association of “another distinct” gut microbiome, *Selenomonadales*, and seleno-compounds with the occurrence of AMI.

However, the relationship between *Selenomonadales* and seleno-compounds needs to be clarified in future studies. We hypothesize that the metabolic pathways in association with *Selenomonadales* and AMI episodes are short-chain fatty acids (SCFAs). Previous studies have demonstrated the gut microbiome-metabolized undigested nutrients to be monosaccharides, which are further fermented to various fatty acids, ranging from 1 to 6 carbon chains, commonly referred to as SCFAs, such as acetate, butyrate, and propionate [15]. About 5% to 10% of SCFAs, particularly butyrates, serve as energy substrates for the epithelial cells of the intestines. Other SCFAs could serve as signaling molecules and modulate several physiological effects associated with cardiovascular diseases (e.g., autonomic systems, blood pressure, immune function, inflammatory responses, and other cellular functions) in addition to energy metabolism [15].

In animal models, SCFAs were found to be critical for repairing the cardiac structure after AMI events [15]. *Selenomonadales* reportedly ferments carbohydrates into the acetate and lactate associated with SCFA formation [16]. Therefore, we hypothesize that these SCFA metabolites generated by *Selenomonadales* might interfere with downstream metabolic pathways and affect coronary atherosclerosis in correlation with AMI episodes. The AMI subjects had less abundant *Selenomonadales*, associated with deprivation of intestinal SCFAs. They might lose the benefits from SCFAs mentioned above. However, this assumption should be clarified in further study.

### 4.3. Selenium in the Pathophysiology of Cardiovascular Diseases

Selenium, first discovered in 1817 by Berzelius, is crucial for various biological functions in the human body, such as thyroid hormone production, antioxidant effects, and the regulation of the autoimmune system [17]. In the serum, selenium is incorporated as the amino acid selenocysteine and carried by selenoproteins to mediate its antioxidant effects [18,19]. Studies show that selenium supplementation could reduce the areas of atherosclerotic lesions in the aorta of rabbits [20,21]. Interestingly, some selenoproteins, such as glutathione peroxidases (GPX1, GPX3, and GPX4), selenoprotein P (SelP), and thioredoxin reductases (TrxR), are abundant within the arterial walls as well [22,23]. Therefore, selenium is closely associated with decreasing atherosclerosis formation.

### 4.4. Mechanisms of Selenium for Preventing Coronary Atherosclerosis

The risk factors traditionally associated with atherosclerosis are diabetes mellitus, hypertension, and hyperlipidemia, all associated with increasing reactive oxygen species (ROS) production in the arterial walls. ROS-related oxidative stress could reduce NO generation; worsen endothelial dysfunction; and induce vascular apoptosis and atherosclerotic plaque ulceration or calcification, all of which are related to AMI [24,25]. Selenoproteins could modulate the ROS signal pathways by inhibiting the nuclear factor-kappa B cascade, reducing the production of interleukins and tumor necrosis factor alpha [26].

Several studies have also revealed that selenium supplementation (most often as sodium selenite) could increase the activity and expression of selenoproteins (GPX1, GPX4, SelP, and TrxR) in vascular endothelia cells or vascular smooth muscle cells. These selenoproteins could inhibit the oxidation of low-density lipoprotein (LDL), cell damage, cell apoptosis, and oxidative stress from the oxidized LDL or oxysterol [27,28,29,30].

In addition to oxidative stress, chronic inflammation, the immune system (monocyte migration, adherence, and phagocytosis of leucocytes), and several adhesion mediators (intercellular adhesion molecule-1 (ICAM-1), vascular cell adhesion molecule-1 (VCAM-1), and E-selectin) are also involved in the atherosclerotic process [31,32]. Supplemental selenium could incorporate into selenoproteins and moderate inflammation; immunity; and the expression of VCAM-1, ICAM-1, and E-selectin via the p38-mitogen-activated protein kinase [33,34].

### 4.5. Association between Selenium and AMI

In the present study, seleno-compounds were noted to be more abundant in the fecal samples obtained from the AMI group after metabolomic analysis. According to previous studies, selenium also affects the gene expression, signaling pathways, microbiome composition, and cellular functions of the intestinal microbiota [35]. As mentioned above, selenoproteins, especially GPX and TrxR, regulate the oxidative balance of target organs within the human body [30,36]. In reaction to AMI events, the levels of selenoproteins in the serum increase and are positively correlated with the severity of the AMI [37,38]. Therefore, selenium administration after myocardial infarction could enhance ROS neutralization and decrease cardiac injury [37]. Previously, serum and urine selenium concentrations in AMI patients have been declared to be significantly lower than those found in the control group [39]. However, in the present study, the seleno-compounds in the feces were more plentiful. All of these taken together lead us to hypothesize that the gut microbiome regulates selenium homeostasis and keeps the body selenium status as high as possible under AMI events [39]. Future studies focused on these selenium profiles could elaborate the definite mechanisms.

## 5. Limitations

Because the inconvenience of feces collection is not well accepted for many patients, only a few patients eventually signed the informed consent form and joined the study. This is the reason why the final case number was low. Additionally, strict exclusion criteria were set up in the present study to exclude the possibly of confounding factors as far as possible. Only 44 participants (19 AMI and 25 control) were left for the final analysis, although 360 patients were initially enrolled. Moreover, some clinical factors were different between the AMI and control patients. We could not perform regression analysis to incorporate these covariates into the linear discriminant analysis. Therefore, our findings should be interpreted carefully to avoid false positive signals and random effects. We are preparing another large sample size study to validate our findings. In future studies, the selenium levels in the serum and feces could be checked as well to reveal more details about the change in seleno-compounds in association with *Selenomonadales* and AMI episodes.

## 6. Conclusions

The present study demonstrated a decreasing abundance of *Selenomonadales* and an increasing abundance of seleno-compounds after AMI. Our findings link gut microbiome and metabolites to AMI. Understanding these relationships could create another dimension of personalized medicine for AMI in the future.

## Figures and Tables

**Figure 1 jcm-11-01462-f001:**
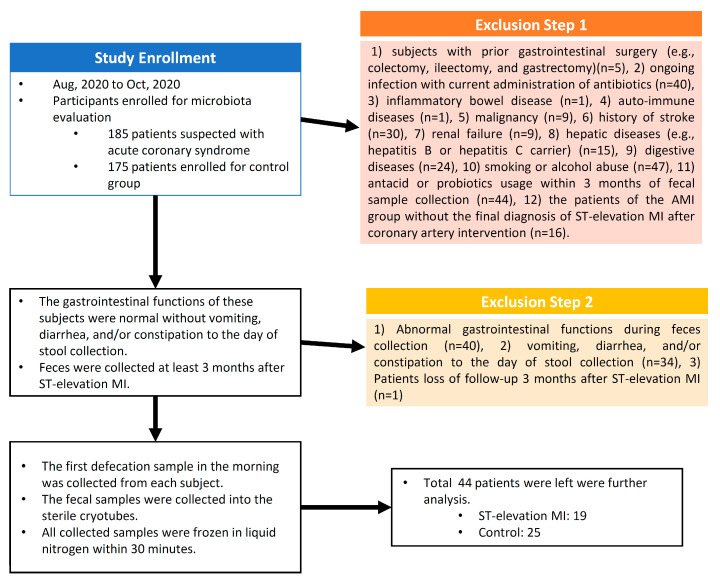
The study flow chart (MI: myocardial infarction).

**Figure 2 jcm-11-01462-f002:**
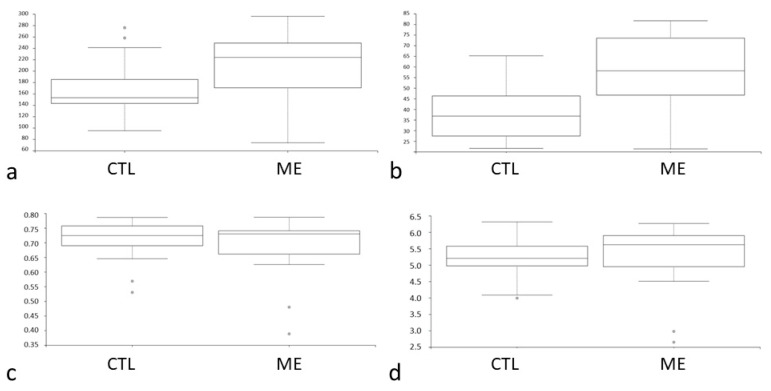
(**a**) Observed operational taxonomic units (OTUs); *p* = 0.01; (**b**) Faith’s phylogenetic diversity (PD); *p* < 0.001; (**c**) Pielou’s evenness; *p* = 0.4; 95% CI; (**d**) Shannon; *p* = 0.23; 95% CI. CTL: control; ME: AMI.

**Figure 3 jcm-11-01462-f003:**
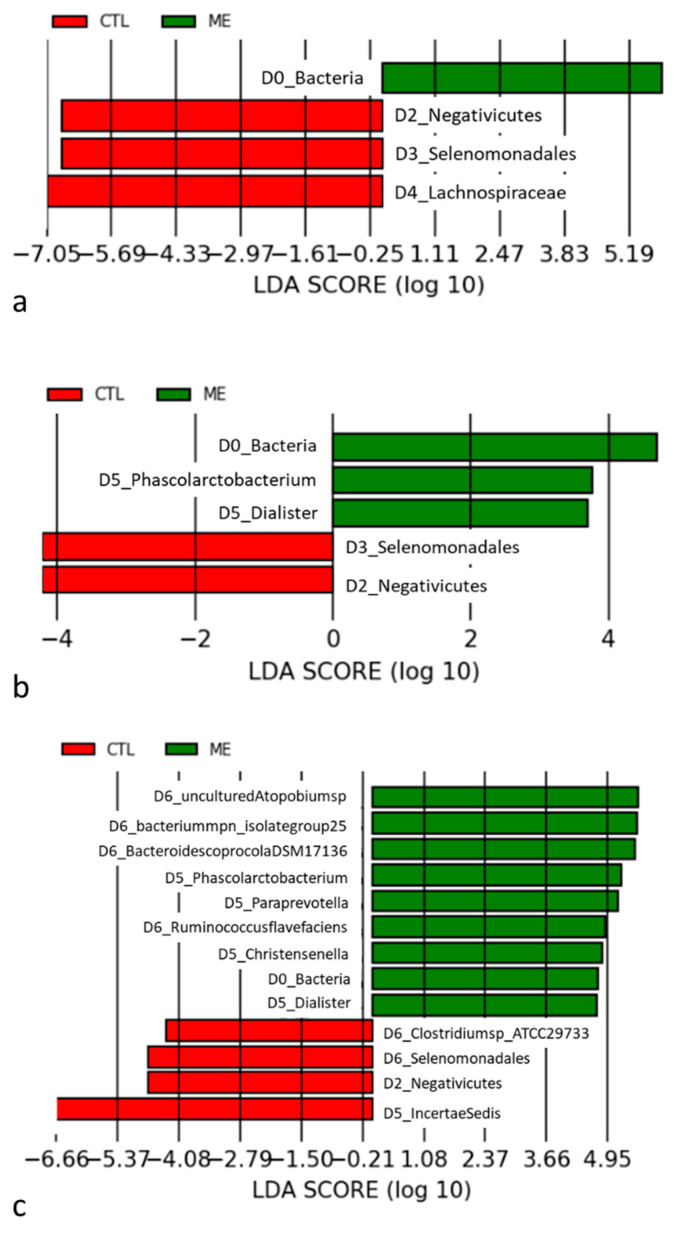
(**a**) Taxonomic groups showing the linear discriminant analysis (LDA) result at the family level; (**b**) taxonomic groups showing the linear discriminant analysis (LDA) result at the genus level; (**c**) taxonomic groups showing the linear discriminant analysis (LDA) result at the species level. CTL: control; ME: AMI.

**Figure 4 jcm-11-01462-f004:**
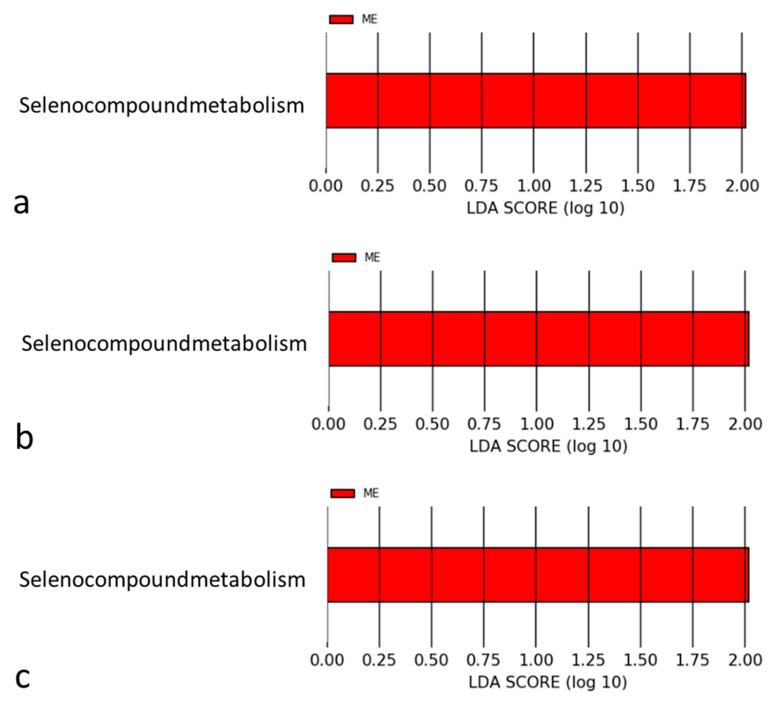
(**a**) Taxonomy differential abundance analysis of seleno-compound metabolism at the family level; (**b**) taxonomy differential abundance analysis of seleno-compound metabolism at the genus level; (**c**) taxonomy differential abundance analysis of seleno-compound metabolism at the species level. CTL: control; ME: AMI.

**Table 1 jcm-11-01462-t001:** Patient characteristics.

	AMI (*n* = 19)	Control (*n* = 25)	*p*-Value
	Mean ± SD	Mean ± SD	95% CI
Age (years)	65.15 ± 11.53	73.20 ± 9.06	<0.001
DM	4 (21%)	3 (12%)	0.43
HTN	14 (74%)	15 (60%)	0.43
Hyperlipidemia	5 (26%)	6 (24%)	0.86
Aspirin	19 (100%)	3 (12%)	<0.001
Plavix	19 (100%)	0 (0%)	<0.001
ACEI/ARB	17 (89%)	14 (56%)	0.01
BB	16 (84%)	1 (4%)	<0.001
BH (cm)	166.08 ± 6.87	158.92 ± 6.62	<0.001
BW (kg)	72.83 ± 13.04	61.08 ± 11.94	<0.001
SBP (mmHg)	137.53 ± 4.45	135.76 ± 3.62	0.35
DBP (mmHg)	85.79 ± 3.16	83.52 ± 2.86	0.83
WBC (10^3^/μL)	7.62 ± 3.28	6.52 ± 1.81	0.14
Hb (g/dL)	13.75 ± 2.57	13.00 ± 2.11	0.23
GLU AC (mg/dL)	114.62 ± 35.73	112.23 ± 36.97	0.81
HbA1c (%)	6.47 ± 1.34	6.23 ± 0.91	0.44
GOT (U/L)	36.46 ± 29.90	23.07 ± 9.13	0.04
GPT (U/L)	26.59 ± 16.43	17.61 ± 10.96	0.02
Bil-T (mg/dL)	0.92 ± 0.46	0.71 ± 0.38	0.22
Bil-D (mg/dL)	0.26 ± 0.26	0.15 ± 0.07	0.37
ALP (U/L)	115.55 ± 152.46	50.33 ± 12.90	0.49
rGT (U/L)	75.82 ± 84.90	17.67 ± 8.50	0.27
BUN (mg/dL)	21.54 ± 12.99	17.96 ± 5.78	0.29
Cr (mg/dL)	1.39 ± 2.08	0.96 ± 0.27	0.32
eGFR (mL/min/1.73 m^2^)	82.79 ± 47.78	79.32 ± 21.79	0.74
UA (mg/dL)	6.29 ± 1.60	5.70 ± 1.42	0.17
Na (mmol/L)	138.23 ± 5.80	138.31 ± 3.79	0.96
K (mmol/L)	4.20 ± 0.44	4.10 ± 0.58	0.46
Ca (mmol/L)	2.29 ± 0.11	2.29 ± 0.17	0.97
Mg (mmol/L)	0.80 ± 0.12	0.90 ± 0.05	0.12
CHO (mg/dL)	151.27 ± 39.44	166.79 ± 44.16	0.16
TG (mg/dL)	126.16 ± 55.33	149.96 ± 111.52	0.27
LDL (mg/dL)	93..06 ± 39.12	94.00 ± 29.19	0.93
HDL (mg/dL)	46.68 ± 12.56	46.70 ± 12.60	0.99
CK (U/L)	895.04 ± 1862.19	105.38 ± 20.33	0.04
CK-MB (U/L)	57.60 ± 118.37	2.69 ± 0.91	0.53
Tn-I (ng/mL)	16.73 ± 21.52	0.01 ± 0.00	0.52
CRP (mg/dL)	3.51 ± 4.59	0.06 ± 0.04	0.03
NT-pro BNP (pg/mL)	1994.68 ± 4037.07	1816.60 ± 3014.64	0.94
LDH (U/L)	221.50 ± 137.27	150.67 ± 52.77	0.41
Lactic acid (mmol/L)	1.56 ± 1.00	0.81 ± 0.16	0.23

CI: confidence interval; AMI: acute myocardial infarction; SD: standard deviation; DM: diabetes mellitus; HTN: hypertension; ACEI/ARB: angiotensin converting enzyme inhibitor/angiotensinogen receptor blocker; BB: beta-blocker; BH: body height; BW: body weight; SBP: systolic blood pressure; DBP: diastolic blood pressure; WBC: white blood cell; Hb: hemoglobulin; GLU AC: fasting blood glucose; HbA1c: glycated hemoglobin; GOT: glutamic oxaloacetic transaminase; GPT: glutamic pyruvic transaminase; Bil-T: total bilirubin; Bil-D: direct bilirubin; ALP: alkaline phosphatase; rGT: r-glutamyl transferase; BUN: blood urea nitrogen; Cr: creatinine; eGFR: estimated glomerular filtration; UA: uric acid; Na: sodium; K: potassium; Ca: calcium; Mg: magnesium; CHO: cholesterol; TG: triglyceride; LDL-c: low-density lipoprotein cholesterol; HDL-c: high-density lipoprotein cholesterol; CK: creatine kinase; CK-MB: creatine kinase MB; Tn-I: troponin-I; CRP: C-reactive protein; NT-ProBNP: N-terminal pro-brain natriuretic peptide; LDH: lactate dehydrogenase.

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
