# Peer review of "The Gut Microbiome, Seleno-Compounds, and Acute Myocardial Infarction"

_jcm, 2022, doi:10.3390/jcm11051462_

Round 1
Reviewer 1 Report
Gut microbiome takes part in many physiologic and pathologic processes. The presented study is aimed to reveal gut micribiome characteristics in patients with acute myocardial infarction. Authors have found 1) abundance of bacteria in patients with MI. 2) Selenomodalens predominance and 3) plentifulness of seleno-compound. Thus, an important component of gut-vessel-heart axis is analyzed. The study is well-done with good laboratory data and statistics. However, several important points have to be discussed and methods and results sections.
- More clinical data have to be added for MI patients: stemi/nonstemi, comorbidities, treatment, complications (congestions? Hypoperfusion?), ICU course etc
- The same about controls. Comorbidities, medication (especially aspirin)
- Timing of stool sampling. First 24 hours after admission or at discharge etc.
- There is not enough data to name the paper as MB as a BIOMARKER of MI. Authors found some changes in microbiota in patients with MI (complicated atherothrombosis and its treatment) in comparations to controls (with/without atherosclerosis etc).
Author Response
Gut microbiome takes part in many physiologic and pathologic processes. The presented study is aimed to reveal gut micribiome characteristics in patients with acute myocardial infarction. Authors have found 1) abundance of bacteria in patients with MI. 2) Selenomodalens predominance and 3) plentifulness of seleno-compound. Thus, an important component of gut-vessel-heart axis is analyzed. The study is well-done with good laboratory data and statistics. However, several important points have to be discussed and methods and results sections.
- More clinical data have to be added for MI patients: stemi/nonstemi, comorbidities, treatment, complications (congestions? Hypoperfusion?), ICU course etc
Response:
Thank you for the reviewer’s precious comments. All of the AMI patients were diagnosed with ST-elevation myocardial infarction. Those AMI patients all accepted percutaneous coronary intervention and following medical treatments at intensive care unit after coronary intervention according to the guideline of American College of Cardiology. [1] No complications such as cardiogenic shock, hypotension, pulmonary congestion, renal or respiratory failure were noted during hospitalization. We added this at the paragraph of “2.2. Study Setting and Participants” of the revised manuscript in red-color characteristics. The baseline clinical data and comorbidities of the AMI patients were also revised at the Table 1.
2. The same about controls. Comorbidities, medication (especially aspirin)
Response:
Thank you for the reviewer’s precious comments. The baseline clinical data and comorbidities of the control patients were also revised at the Table 1.
3. Timing of stool sampling. First 24 hours after admission or at discharge etc.
Response:
Thank you for the reviewer’s precious comments. To minimize the potential confounding effects of antibiotics on the microbiota, it was ensured that none of the participants were given any antibiotics for at least 3 months after AMI or enrollment and the feces were collected at least 3 months after AMI. We added this at the paragraph of “2.3. Fecal Collection and Processing” of the revised manuscript in red-color characteristics.
4. There is not enough data to name the paper as MB (microbiota) as a BIOMARKER of MI. Authors found some changes in microbiota in patients with MI (complicated atherothrombosis and its treatment) in comparations to controls (with/without atherosclerosis etc).
Response:
Thank you for the reviewer’s precious comments. We had revised the title and the introduction as the reviewer’s suggestion accordingly.
Reviewer 2 Report
The basic limitation of the study is the small size of the study group. Only 44 patients were studied, and 19 patients with myocardial infarction (MI) were compared with 25 controls. The results may be random. It is necessary to perform the study on a larger group of patients, at least 30 with MI vs. 30 controls.
Other disadvantages of the manuscript:
1. The purpose of the research should be separated as a subsection of the introduction.
2. The exclusion criteria are poorly described. How many people were excluded from the study? How many based on specific criteria?
3. The description of the statistical analysis is too basic. The rationale for the selected tests must be added.
4. A number of clinical parameters differed in the study groups (see Table 1). How was the influence of these differences on the observed relationships assessed? Perhaps you should use regression analysis.
5. The compared groups differed in such basic variables as age and BMI. You may need to use the case to case method in selecting a control group. Then you will avoid the problem of the influence of age and weight on the observed relationships.
6. Study limitations were not discussed.
7. Only 21 references were cited. There is a lack of much current articles on the role of selenium in the pathophysiology of cardiovascular diseases.
Author Response
The basic limitation of the study is the small size of the study group. Only 44 patients were studied, and 19 patients with myocardial infarction (MI) were compared with 25 controls. The results may be random. It is necessary to perform the study on a larger group of patients, at least 30 with MI vs. 30 controls.
Response:
Thank you for the reviewer’s precious comments. We had revised the paragraph of “5. LIMITATIONS” with the reviewer’s reminding.
Other disadvantages of the manuscript:
The purpose of the research should be separated as a subsection of the introduction.
Response:
Thank you for the reviewer’s precious comments. The purpose of this research has been revised as a subsection of the introduction.
The exclusion criteria are poorly described. How many people were excluded from the study? How many based on specific criteria?
Response:
Thank you for the reviewer’s precious comments. The study flow chart with the specific inclusion/ exclusion criteria was added as the Figure 1 and the paragraph of “ 2.2. Study Setting and Participants” has been revised in red-color characteristics.
The description of the statistical analysis is too basic. The rationale for the selected tests must be added.
Response:
Thank you for the reviewer’s precious comments. The paragraph of “2.7. Statistical Analyses” has been revised in red-color characteristics.
A number of clinical parameters differed in the study groups (see Table 1). How was the influence of these differences on the observed relationships assessed? Perhaps you should use regression analysis.
Response:
Thank you for the reviewer’s precious comments. Some clinical factors were different between AMI and control patients. We could not perform regression analysis to incorporate these covariates into the linear discriminant analysis. We added this as a limitation in the manuscript.
The compared groups differed in such basic variables as age and BMI. You may need to use the case to case method in selecting a control group. Then you will avoid the problem of the influence of age and weight on the observed relationships.
Response:
Thank you for the reviewer’s precious comments. As this was a study with small case numbers left for final analysis and some clinical factors were different between AMI patients and control patients, we used another nonparametric analysis to compare. The finial results were the same as the previously observed.
Study limitations were not discussed.
Response:
Thank you for the reviewer’s precious comments. We had revised the paragraph of “5. LIMITATIONS” in red-color characteristics.
Only 21 references were cited. There is a lack of much current articles on the role of selenium in the pathophysiology of cardiovascular diseases.
Response:
Thank you for the reviewer’s precious comments. The role of selenium in the pathophysiology of cardiovascular diseases has been discussed at the paragraph of “4. DISCUSSION” in red-color characteristics.
Round 2
Reviewer 1 Report
The article could be published
Author Response
Reviewer 1
The discussed limitations of the paper have been more clearly stated in the revised version. The flow chart (Figure 1) still needs to be revised. In the current version, there is no clear information about how many subjects were excluded and for what reasons. Specifically, for Exclusion Step 1 and Exclusion Step 2, the individual reasons are listed, but no frequencies are given for the individual reasons. After the revision of Figure 1, the mansucript can be accepted for publication.
Response:
Thank you for the reviewer’s precious suggestions and the Figure 1 has been revised accordingly.